# The P.E.A.C.E. Pack Program in Italian High Schools: An Intervention for Victims of Bullying

**DOI:** 10.3390/ijerph17145162

**Published:** 2020-07-17

**Authors:** Annalisa Guarini, Laura Menabò, Damiano Menin, Consuelo Mameli, Grace Skrzypiec, Phillip Slee, Antonella Brighi

**Affiliations:** 1Department of Psychology, University of Bologna, 40127 Bologna, Italy; laura.menabo@unibo.it; 2Faculty of Education, Free University of Bolzano, 39042 Bressanone, Italy; damiano.menin@unibz.it (D.M.); antonella.brighi@unibz.it (A.B.); 3Department of Educational Sciences, University of Bologna, 40126 Bologna, Italy; consuelo.mameli@unibo.it; 4College of Education Psychology and Social Work, Flinders University, South Australia, 5001 Adelaide, Australia; grace.skrzypiec@flinders.edu.au (G.S.); phillip.slee@flinders.edu.au (P.S.)

**Keywords:** bullying, victimization, teacher, adolescents, self-efficacy

## Abstract

*Background:* Bullying is a serious public issue, which mainly occurs in school with negative consequences for the students involved as victims. Very few teacher-delivered interventions have shown positive changes in the victims. The present study aimed at implementing the P.E.A.C.E. (Preparation, Education. Action, Coping, Evaluation) pack program, developed in Australia, in Italian high schools. *Method:* The effectiveness of the program was analyzed through an observational study (pre/post-intervention), involving 551 Italian high school students who completed a questionnaire on bullying victimization, self-efficacy, and bystander behavior. The students were divided into three groups (not involved students, occasional and severe victims) according to their self-reported victimization in the pre-intervention. *Results:* After the intervention, severe victims (victimized once/week or more often) showed a significant decrease in victimization and higher scores in self-efficacy, while an increase in victimization was observed in the not involved students. As reported by all the groups after the intervention, classmates were perceived more likely to intervene when a bullying episode occurred. By contrast, occasional and severe victims perceived their teachers as less likely to intervene. *Conclusions:* The P.E.A.C.E. pack is a promising program confirming in Italian schools the effectiveness already shown in other countries. This program is very useful for severe victims, supporting their self-confidence with a decrease in the frequency of aggressive episodes.

## 1. Introduction

Bullying is a widespread problem [1], with prevalence rates estimated around 35% for bullying perpetration and 36% for bullying victimization [2]. Large-scale cross-national surveys have confirmed the significant extent of bullying. For instance, the international HBSC (Health Behavior in School-aged Children) study [3], including 44 countries across Europe and North America, found a percentage rate of serious victimization (i.e., two or three times a month in the last two months) of about 12% in 13-year-old students. In relation to sequelae, bullying continues to be a severe public health issue, with victims commonly showing symptoms of depression and anxiety [4]. In particular, relational victimization has been found to have a negative effect on social relationships and internalizing problems [4]. Furthermore, bullying is a risk factor for suicide attempts among adolescents globally, as revealed by a recent study involving 48 countries [5], which showed that the prevalence of suicide attempts increased as a function of the frequency of bullying in the past month.

Given the significance of the bullying phenomenon in society, it is not surprising that many scholars have dedicated their efforts to setting up and evaluating the effectiveness of anti-bullying interventions (for a meta-analytical review [6]).

### 1.1. Teacher-Led Anti-Bullying Programs

Notwithstanding that most anti-bullying programs are commonly conducted in the school context, since it is precisely in school that most bullying episodes occur, many of them rely on professionals external to the school to deliver the intervention. Indeed, only a few teacher-led interventions, showing contrasting results, have been reported in the literature [7]. An example of a teacher-based anti-bullying program with promising results was the BIAS (Bullying in Sicilian Schools) study [8], implemented by the teachers of the first grade of secondary schools, revealing a decrease in the number of bullying episodes among students. In contrast, the SEHER (Strengthening the Evidence base on effective scHool based intErventions for pRomoting adolescent health.) intervention, carried out with students aged 13–14 years, showed a significant decrease in bullying and victimization only when delivered by lay counselors, and no evidence for the efficacy of the intervention was found when it was delivered by teachers [9]. A different trend emerged from the ViSC (Viennese Social Competence) Program delivered by teachers, which has been effective in reducing victimization, while no significant effect on reducing aggressive behaviors was reported [10]. Further support for the effectiveness of interventions entirely delivered by teachers were found in cyberbullying studies, revealing a decrease in cyber aggression (Asegúrate Program [11]; Media Heroes [12]) as well as an increase in student coping strategies (RPC, Relazioni per crescere—Relationships to Grow [13]).

Despite these mixed findings, the central role of teachers in tackling bullying has been highlighted in several countries. For instance, in Italy, the guidelines provided by the Ministry of Education, Universities and Research [14] underlined the need for teacher training in order to detect the signs of bullying and manage bullying episodes. In the Australian context, the Australian Wellbeing Framework (See, for example, https://studentwellbeinghub.edu.au/) currently promotes and supports the training of teachers to address school bullying.

Teacher-driven interventions present two main advantages. First, they have a low cost in terms of human and economic resources in comparison with externally delivered programs by psychologists or social workers, and this is a significant funding factor considering the lack of financial resources in low- and middle-income countries [15]. Indeed, the cost–benefit analysis of anti-bullying programs is crucial in convincing policy makers and practitioners to implement interventions [16]. Second, involving teachers in training paths focused on the bullying phenomena and in delivering anti-bullying programs for students could potentially strengthen their ability to directly deal with bullying in their everyday classroom practices [7,17]. This may foster teachers’ self-efficacy [7] that in turn may positively impact on their recruitment and retention, and more broadly on improvements in self-esteem, or good citizenship among students [18]. However, teacher-driven interventions could also have some limitations. Relying on large amounts of “donated” teacher time may have adverse effects on teacher efficacy with regard to a program’s implementation [19]. Indeed, the interventions delivered by teachers may compete with teachers’ duties, especially in the context in which teachers are overloaded [9].

### 1.2. Effectiveness of Anti-Bullying Interventions on Victims

As suggested by Nickerson [20] in her recent review, most anti-bullying interventions have been carried out following universal prevention approaches. However, Kaufman and colleagues [21] pointed out that even if universal anti-bullying interventions can produce effective results, some children with high levels of peer rejection and individual risk factors (e.g., internalizing and parent–child relationship problems) continue to be victimized. In addition, Rigby [22] stressed that when bullying is particularly severe, teacher-led interventions were less successful.

To overcome the possible low effectiveness of teacher-led classroom-based interventions for severe victims of bullying, two approaches are possible. On the one hand, it is possible to offer interventions, which are solely targeted at seriously victimized students as in the following example. The Cognitive Behavioral Group Therapy, administered by two trained social workers and delivered to a selected sample of pure victims, revealed a decrease in physical, verbal and social forms of victimization, as well as in the level of anxiety and depression across three longitudinal time point measurements [23]. The Social Skills Training, developed for the victims of bullying, revealed an increase in their self-esteem, but not improvements in victim status [24]. Although these kinds of interventions may be useful for victims, they do not act on the context in which the students are placed. On the other hand, other programs pointed out a positive impact on the victims by paying particular attention to improving the interpersonal context, involving the whole class and promoting the social integration of rejected students [25].

### 1.3. The Present Study: An Adaptation of the P.E.A.C.E. Pack Program in Italy

As highlighted, although most anti-bullying interventions are carried out in the school setting, only a few of them are conducted by teachers. In addition, many universally based interventions have shown unsatisfactory results for students being seriously bullied. In light of these findings, in this study we present the results of a teacher-led school-based anti-bullying intervention with a specific focus on the effect of this program on victims.

The P.E.A.C.E. (Preparation, Education, Action, Coping, Evaluation) pack program (see the web page www.caper.com.au) [26], which is currently in its fourth edition, represents a framework for schools and teachers to assist them in implementing an anti-bullying program in the school and classroom. The P.E.A.C.E. pack program—which has been developed in collaboration with teachers, students, principals, parents and school administrators—relies on positive psychology theories of change [27]. Positive psychology interventions have been characterized [28] as “programs, practices, treatment methods or activities aimed at cultivating positive feelings, positive behaviors, or positive cognitions” (p. 467). In this way, positive psychology interventions distinguish themselves from wellbeing initiatives such as “pure” anti-bullying programs, ‘quit smoking’ programs, and depression-reducing programs that seek to enhance wellbeing through the removal or reduction of negative factors. The P.E.A.C.E. pack lesson content is integrated with key elements of social and emotional competencies, e.g., relationship skills, and promotes positive attitudes and positive behaviors rather than focusing only on negative behaviors. For this reason, the P.E.A.C.E. pack involves six lessons and activities that focus on the promotion of healthy attitudes such as empathy, optimism, resilience, conflict resolution strategies, kindness and positive emotions among classmates. This positive approach is particularly useful for vulnerable students [27], such as those who have been bullied, improving the interpersonal context and promoting social integration. The main topics of the six lessons include:

*Lesson 1:* “Fostering friendship”. A group-work activity on the meaning of friendship is introduced by asking students some questions such as “*What are the advantages of having friends?*”; “*What are the ways to make new friends?*”.

*Lesson 2:* “Coping with verbal bullying and fostering resilience”. A short video about a girl who was verbally bullied by classmates serves as input for a group discussion about the impact of the insults and the difference between a joke and an insult. The second part of the lesson is aimed at developing the students’ resilience in the face of bullying.

*Lesson 3:* “Cyberbullying-Self concept”. A short video on cyberbullying invites students to reflect on the feelings of the victim and how teachers and parents might be able to help. The second part is aimed at improving the students’ self-image.

*Lesson 4:* “Exclusion-Optimism”. A short video helps the students to focus on the different types of exclusion and how to handle such experiences. In the second part, the aim is on adopting an optimistic attitude.

*Lesson 5:* “Physical Bullying-Conflict resolution”. Starting from a short video about physical bullying, students can discuss different ways to deal with it, adopting both the victim’s and the bystander’s points of view. In the second part, the focus is on improving adaptive conflict management and problem-solving strategies.

*Lesson 6:* “Positive Class Climate”. Group activities are proposed to improve a positive class climate and positive relationships among classmates.

The P.E.A.C.E. pack program is a detailed, multi-lesson, multi-media school-based intervention, with activities and lesson modules that are developmentally appropriate [29]. The intervention is manualized in order to support teachers in the implementation of activities in their classes. The evaluations involve pre- and post-testing. In the fourth edition of the P.E.A.C.E. pack, particular attention is given to assessing the implementation quality of the interventions using an online ‘implementation index’, assessing the fidelity, dosage and quality of the program that is completed by the teachers [30].

This program has been evaluated in primary and secondary schools in Australia (over 90 schools) [31] and translated and implemented in Japan, Malta and Greece [32]. The evaluations have involved pre- and post-testing and longitudinal studies during 1–3 years [33]. Over 12,000 students and their teachers have been involved in the evaluations. The studies reported significant reductions in self-reported victimization, especially among severely victimized students (17–33%), reductions in the bullying of other students (5–10%), increases in coping skills, feelings of safety at school, school belonging and benefits to the reported wellbeing of students [30].

### 1.4. Aims

A priority for reducing bullying is to provide schools with evidence-based, manageable and cost-effective anti-bullying interventions that can be delivered by teachers. It is hypothesized that these interventions should prove to be most effective for the most vulnerable students, who may also be the most resistant to change [21]. Thus, the present study aimed at analyzing, for the first time in Italy, the effect of the school-based P.E.A.C.E. pack intervention, implemented by trained teachers, among high-school students. We focused on high-school students since the evidence of the effectiveness of anti-bullying programs are less consistent in this age group [16,34,35]. We hypothesized stronger effects for severe victims compared to the occasional victims and the not involved students. This is consistent with the adaptation of the P.E.A.C.E. pack in Maltese schools [25], with a reported reduction in the frequency of victimization and an increase in self-efficacy of dealing with bullying and in the positive perceptions of bystanders intervening in anti-bullying episodes. We hypothesized stronger effects for severe victims, since these students may not only have higher levels of victimization before the intervention, but also lower scores in self-efficacy and poorer perceptions of support from bystanders (classmates and teachers) compared to their peers.

## 2. Methods

### 2.1. Participants

To conduct the present study, we made use of a convenience sample methodology with teachers participating autonomously and selecting the classes in which to implement the P.E.A.C.E. pack program. The teachers’ recruitment and data collection took place in the school year 2018/2019. An invitation to participate in the P.E.A.C.E. pack teacher-training course was sent by the Regional School Office of the Italian Ministry of Education to all the high-school teachers of nine provinces in the Emilia-Romagna region. Furthermore, the same invitation was extended to high-school teachers of one province in the Veneto Region. Of the 105 places made available for the training course, 88 teachers from Emilia-Romagna and 15 from Veneto volunteered to participate. As illustrated in the flow diagram of recruitment and selection of teachers (see Figure 1), of the 103 initial participants, only 43 (42%) actually took part in the study (providing both the intervention and the administration of the pre- and post-intervention questionnaires), and the data from their students were used in this study.

In the pre-intervention phase, 862 students completed the questionnaire, while 814 participated in the post-intervention phase. The different number of participants in the first and second wave of data collection depended on who was present at school on the day the pre- and post-questionnaires were administered. Overall, 551 students from 23 high schools (458 from Emilia Romagna and 93 from Veneto) completed the questionnaires with correct matched personal nicknames, and were thus included in the final data set for the analysis. The sample comprised 60% (*n* = 331) males and 40% (*n* = 220) females. In relation to year levels, 54% (*n* = 299) were students in their first year of high school, 33% (*n* = 181) were in their second year, 12.5% (*n* = 68) in their third year and 0.5% (*n* = 3) in their fourth year. Student ages ranged from 13 to 17 years (*M* = 14.8, *SD* = 0.9). Most (90%, *n* = 495) students were born in Italy. In relation to the parental level of education, 19% (*n =* 102) of fathers and 24% (130) of mothers had a university degree; 57% (*n =* 312) of fathers and 60% (*n =* 331) of mothers had a diploma from high school; while 24% (*n =* 132) of fathers and 16% (*n =* 87) of mothers had obtained a lower secondary school or primary school certificate.

### 2.2. Questionnaire

A questionnaire consisting of four different sections was delivered to all students.

#### 2.2.1. Bullying Victimization

To assess the level of bullying victimization, the European Bullying Intervention Project Questionnaire was administered (ECIPQ; [36,37]). The ECIPQ is a 7-item self-report questionnaire aimed at investigating the level of victimization (e.g., “*Someone hit, kicked, or pushed me*”, “*Someone spread rumors about me*”). It is based on a 5-point Likert-type scale (0 = no; 1 = yes, once or twice; 2 = yes, once or twice a month; 3 = yes, about once a week; and 4 = yes, more than once a week). Cronbach’s alpha was 0.70, showing satisfactory reliability.

#### 2.2.2. Self-Efficacy

To measure the student self-efficacy in dealing with bullying, an 11-item scale was developed and adapted from Bandura [38]. The items focused on the students’ perceived ability to handle bullying (e.g., “*I am able to defend myself*”, “*I can make other people stop when they make fun of me or bother me*”). Students provided responses using a 6-point Likert scale of agreement/disagreement (0 = completely disagree; 1 = disagree; 2 = slightly disagree; 3 = slightly agree; 4 = agree; 5 = completely agree). Cronbach’s alpha was 0.86, showing good reliability.

#### 2.2.3. Bystander Behavior

To investigate the perception of bystanders’ behavior in intervening to stop bullying when students or teachers witnessed it, two items were administered (i.e., “*What do teachers at school usually do when they see it?*” “*What do students at school usually do when they see it?*” [30]). Participants were asked to indicate their response on a 4-point Likert-type scale (1 = they always try to stop it; 2 = they sometimes try to stop it; 3 = they barely try to stop it; 4 = they never try to stop it).

#### 2.2.4. Demographic Information

A section regarding the demographic information (age, gender, parental level of education, place of birth) was included at the end of the questionnaire.

### 2.3. Procedure and Study Design

The volunteer teachers participating in this program attended a professional development-training course lasting 6 h. The training, carried out by expert psychologists, was aimed at introducing the contents of the P.E.A.C.E. pack and its goals, and provided detailed information on its procedures and implementation (i.e., classroom activities for each of the six lessons; see the web page www.caper.com.au). In addition, each teacher was provided with a training manual [39] containing step-by-step descriptions of the activities. The proposed activities/lessons were six in number, each lasting 1.5/2 h (as described in the introduction). All the activities proposed for the implementation of the P.E.A.C.E. pack were run by classroom teachers within a two-month period.

In order to assess the effectiveness of the intervention, a repeated measures design was adopted. Students completed online questionnaires during school hours using computers in multimedia classrooms before the intervention (pre-intervention) and immediately at the end (the last lesson) of the intervention (post-intervention). Teachers remained in the classroom during the data collection in order to clarify any questions or doubts. Questionnaires were anonymous as a personal nickname was chosen by each student and used for the two waves of data collection (pre-intervention and post-intervention) in order to match their pre- and post-responses.

Teachers who administered the program completed an online questionnaire to assess the fidelity (5 questions, e.g., “*I was faithful in teaching the lessons as suggested*”), dosage (5 questions, e.g., “*All lessons were delivered in the time required*”) and quality (5 questions, e.g., “*Students enjoyed the P.E.A.C.E. pack lessons*”) of their program delivery. Teachers provided responses according to a 6-point Likert scale of agreement/disagreement (1 = very strongly disagree to 6 = very strongly agree). Mean scores were calculated for each dimension (dosage, *M* = 4.5, *SD* = 0.8; quality, *M* = 4.6, *SD* = 0.6; fidelity, *M* = 4.2, *SD* = 0.7).

### 2.4. Ethics

The formal approval for the study was provided by the Bioethics Committee, University of Bologna (prot. no. 17372). Since the participants were minors, both parents gave their informed written consent for their child’s participation in the study. We also asked the students for written informed consent, according to the Declaration of Helsinki. In the information statement, the participants were informed about the purpose of the research and the procedures; the benefits/risks of participating in this study; the rights to decline to participate and to withdraw from the research without consequences; and the contact details of the researchers for any questions. In the present project, no incentives or benefits for participation were provided.

### 2.5. Statistical Analysis

Data from the pre-intervention questionnaire were used to classify the students according to their victimization score. We coded the students with a score on the ECIPQ between 0 and 1 as “not involved”, students with a score between 2 and 5 as “occasional victims”, and the students with a score in the ECIPQ ≥ 6 in the last two months (corresponding to three times a month in the last two months) as “severe victims”. In our sample, 45% (*n* = 250) of the students were classified as not involved in bullying victimization, while over one in two students (55%, *n* = 301) reported being bullied. Of these, 13% (*n* = 70) were classified as severe victims and 42% (*n* = 231) as occasional victims. We analyzed the differences among the three groups in their pre-intervention self-efficacy and their perceived bystander behavior (students and teachers) with Kruskal–Wallis (H) tests as the data were skewed. Eta-squared (*η*^2^*_H_*) was computed as a measure of effect size where from *η*^2^*_H_* = 0.010 to 0.059 represents a small effect; from *η*^2^*_H_* = 0.060 to 0.110 represents a medium effect; and *η*^2^*_H_* > 0.140 represents a large effect [40]. Post-hoc Mann–Whitney tests were run using a Bonferroni alpha level of 0.017 (0.05/3). The correlation coefficient *r* was computed as an effect size estimation where *r* = 0.10–0.30 represents a small effect; r = 0.30–0.50 represents a medium effect; and r > 0.50 represents a large effect [41,42]. Wilcoxon matched-pair tests were carried to analyze the differences between the pre- and post- interventions in bullying victimization, self-efficacy in dealing with bullying, and the perceived bystander behavior (students and teachers). Correlation coefficient *r* was computed as the effect size estimation. The statistical analyses were undertaken using IBM SPSS Statistics 25 (IBM Corp., Armonk, NY, USA).

## 3. Results

### 3.1. Bullying Victimization

Regarding the effect of the intervention, the Wilcoxon matched-pair tests showed a significant increase in bullying victimization in not involved students (Table 1), no significant difference in occasional victims (Table 2), and a decrease in severe victims (Table 3).

### 3.2. Self-Efficacy in Dealing with Bullying

A Kruskal–Wallis test revealed a significant difference in self-efficacy between the three groups at the pre-intervention phase, *H*(2) = 62.60, *η*^2^*_H_* = 0.12, *p* < 0.001, with lower scores among the severe victims, compared to the occasional victims (*U* = 4894, *r* = 0.29, *p* < 0.001) and the not involved students (*U* = 3619, *r* = 0.42, *p* < 0.001). In addition, occasional victims showed lower self-efficacy scores compared to the not involved students (*U* = 22186, *r* = 0.20, *p* < 0.001).

In relation to the effect of the intervention, a significant increase in self-efficacy was found for the severe victims at the end of the intervention (Table 3). However, no difference in the self-efficacy was observed between the pre- and post-intervention among the not involved students and occasional victims (Table 1 and Table 2).

### 3.3. Bystander Behavior

A Kruskal–Wallis test revealed that the not involved students, the occasional and the severe victims differed in their perception of the bystanders’ behavior of their classmates pre-intervention, *H*(2) = 34.62, *η*^2^*_H_* = 0.07, *p* < 0.001. Severe victims perceived their classmates as less likely to intervene compared with occasional victims (*U* = 5896, *r* = 0.21, *p* < 0.001) and the not involved students (*U* = 5016.5, *r* = 0.32, *p* < 0.001). In addition, the occasional victims perceived that there was less willingness to intervene by classmates compared to the not involved students (*U* = 23835.5, *r =* 0.14, *p* = 0.002). The same was true for the perceived teacher concern, *H*(2) = 24.07, *η*^2^*_H_* =0.04, *p* <0.001. Severe victims perceived their teachers as less likely to intervene compared to occasional victims (*U* = 6462.5, *r* = 0.15, *p* < 0.001) and the not involved students (*U* = 5683.5, *r* = 0.27, *p* < 0.001); occasional victims perceived less intervention by teachers compared to the not involved students (*U* = 23642.5, *r* = 0.14, *p* = 0.003).

A Wilcoxon matched-pair test of the perception of the bystanders’ behavior to intervene in bullying revealed that, after the intervention, all the students perceived their classmates as more likely to intervene when a bullying episode occurred. Indeed, this result was significant for the not involved students (Table 1), occasional victims (Table 2) and the severe victims (Table 3). In contrast, the perception of the teacher bystander behavior changed differently among the groups. Indeed, after the intervention, both the not involved students (Table 1) and occasional victims (Table 2) perceived their teachers as less likely to intervene when a bullying episode occurs in class. At the same time, no change was reported by the severe victims (Table 3).

## 4. Discussion

The P.E.A.C.E. pack is an international intervention program addressing bullying in schools. It provides school-based strategies delivered in the form of discrete lessons by classroom teachers that have been shown to reduce school bullying [30]. The present study describes the findings from the Italian adaptation of the P.E.A.C.E. pack intervention, implemented in a range of high schools in Northern Italy.

The findings from the present study confirm that school bullying is a frequent aspect of Italian students’ lives, with approximately one in two reporting being bullied and over one in ten reporting that they were seriously bullied. The incidence of severely bullied students was in line with other international reports [3], while the incidence of victimization was higher in our study than in other studies [2,43,44]. This result can be explained by the fact that we have decided to include all those students who reported having suffered at least two episodes of bullying in the last two months as occasional victims, using a broad criterion to include all the victimized students.

The analysis concerning those students identified as severely bullied before the intervention program suggests that the intervention had a significant and positive impact on them, reducing the level of their self-reported victimization in the post-intervention phase. Similar results were described in the adaptation of the P.E.A.C.E. pack in Maltese schools that showed a significant reduction in the level of self-reported bullying among severely bullied students [25]. The effectiveness of the intervention for severe victims, even with a small effect size is particularly relevant since previous studies have showed that children with high levels of peer rejection may continue to be victimized after anti-bullying interventions [21], especially when the interventions were delivered by teachers [22]. In addition, our study suggested that it is possible to intervene not only in relation to severe victims [23,24], but to include the whole class in the program. As such, the cascade effect of even a small effect size should be considered as a positive outcome in light of changing the classroom climate to a more positive and welcoming one for students.

In contrast, we found an increase in victimization with a medium effect size among the not involved students. As suggested by Ttofi and Farrington’s meta-analysis [16], when anti-bullying programs include working with peers (e.g., encouraging bystander intervention to prevent bullying), as in the P.E.A.C.E. pack program, an increase in victimization may occur. However, as described in the review of the research earlier in this paper [44], the outcomes of a range of anti-bullying intervention programs report mixed findings with up to 30% of the studies showing no significant reductions in victimization. The result of our study should also be interpreted in light of the cost–benefit ratio. Indeed, as suggested by reviews and meta-analyses, the intervention programs best show their positive effects when they are intensive and long-lasting [16,45]. The P.E.A.C.E. pack intervention is intensive, with the great advantage of having a low cost. Further studies are needed to evaluate whether, by increasing the duration of the intervention, positive results could be achieved, particularly for the not involved students.

A second relevant and encouraging medium effect size finding of our study regards the increase in student self-efficacy in dealing with bullying amongst the seriously bullied students. These data support the utility of the P.E.A.C.E. pack program, as the literature highlights that the coping resources of children, and consequently their perception of self-efficacy, may be severely taxed by the repeated experiences of victimization [46]. Indeed, bullying incidents, which are frequent and occur over long periods of time, may overwhelm student’s self-efficacy for coping. Improving self-efficacy in dealing with bullying among severely victimized adolescents may foster a crucial protective factor such as resilience [47,48]. As Bandura [49] pointed out, the most powerful work in resilience is promoting personal agency and confidence in one’s ability to develop resilience. Self-efficacy in dealing with bullying might influence the actions and behaviors that can reduce the risk of being further victimized, and at the same time, reduce the learned helplessness that severe victims may experience. According to these suggestions, even a small effect of self-efficacy can result in a cascade effect on resiliency and on emotional symptoms experienced by severe victims, as suggested for the victims of cyberbullying [48]. This is an interesting and innovative finding, with theoretical and practical implications for school-based interventions. Concerning the occasional victims and the students who were not involved in victimization, we did not find an effect of the intervention on their self-efficacy. One would not expect a great deal of change in self-efficacy amongst these students, since their self-efficacy was less compromised, as shown by their average scores in the pre-intervention assessment.

A third relevant finding in the present study is an encouraging improvement (medium effect size) in the severe victims’ perceptions of bystanders’ actions to counteract bullying, revealing that after the intervention, they perceived their classmates as more likely to intervene when a bullying episode occurred. The same trend was described in the students not involved in victimization, as well as in occasional victims, even if the effect sizes were small. The difference in the magnitude of the effect size can be explained by the fact that severe victims perceived fewer actions by their classmates before the intervention, compared to the occasional victims and not involved students. Thus, the P.E.A.C.E. pack intervention was not only effective for severe victims, but the findings suggest that all the students recognized a greater willingness in their classmates to accept some social responsibility for supporting their fellow students who may be experiencing difficulties in terms of bullying. Such social accountability is particularly relevant because, as revealed by other studies, many victims do not seek help for the fear of retaliation from bullies and shame over peers’ perceptions of them [50,51]. Indeed, around 30% of bullied pupils in English schools told no one [52], and it was atypical for Australian students to report bullying to school counsellors or adults [53]. For these reasons, an important element of many anti-bullying programs is encouraging bystanders to intervene [54]. Indeed, this is the number one coping strategy reported by trainee teachers as the tactic they would most recommend to students [55,56].

Interestingly, a different trend was found in relation to the effect of the intervention on the perceived teachers’ willingness to intervene according to the students. Indeed, contrary to the effect described for the classmates’ bystander behavior, after the intervention, the students not involved in victimization and occasional victims perceived their teachers as less likely to intervene when a bullying episode occurred; whereas severe victims did not detect any difference in the teachers’ behavior. One possible explanation is that students do not “see” the positive things that teachers do to make the classroom and school safe from bullying. However, other interpretations are also possible. First, we should consider that students were asked about their perceptions of teachers in general, and not specifically about those teachers involved in delivering the program. It is therefore possible that, in general, the teachers’ behavior did not actually change before and after the intervention in the students’ view. It is also necessary to consider that most bullying episodes occur in places at school that are not always controlled by teachers, such as bathrooms or corridors, and this makes direct and timely teacher intervention more difficult. Furthermore, as already described in other contexts [9], when teachers are overloaded, interventions in relation to bullying may be hindered or neglected as they compete with other teachers’ duties related to instruction and academic achievement. Nevertheless, the literature converges in highlighting that students’ socio-emotional wellbeing and school performance are far from being independent [57]. Consequently, in our view, the teacher’s role cannot be limited to the transmission of information and knowledge, but should also include the ability to consider and to deal with the relational and emotional dynamics emerging in their classes every day. Based on these considerations, our study seems to emphasize the importance of making teachers’ actions more frequent, effective and visible for pupils, since it is very crucial that teachers can be a trusted point of reference for dealing with bullying, especially for victims [57]. In addition, as revealed by a recent meta-analysis [7], although teachers are involved in the bullying phenomena, most studies have not analyzed the possible effects of anti-bullying programs on teachers’ intervention practices. However, even if anti-bullying programs can improve teachers’ self-efficacy in dealing with bullying, as well as their knowledge of the phenomenon, when students’ perceptions of their teachers in dealing with bullying have been analyzed [7], only small effects have been found.

### Limitations and Strengths

The research from the pilot study in the Italian schools reported here has a number of limitations. A first limitation was the design of this study, which was carried out through the analysis of change between pre- and post-intervention phases, without including a control group. A further study, including a control sample of students who were bullied but did not receive the intervention, could help to better understand the influence of the P.E.A.C.E. pack program on students. This would allow an account of maturation effects. However, this was not possible in the present study due to the schools’ requests. Furthermore, there are ethical issues associated with withholding a possibly effective intervention from a body of students. In addition, a follow-up measure could help to understand if the effects described after the intervention persist over time. Limitations also included the use of self-reported data by students in relation to school bullying. In further studies, teachers could also be included in the evaluation of the level of bullying. Furthermore, the data on school performance could be useful in understanding the possible effect of the intervention on academic achievement, assuming a positive association between students’ social-emotional skills and academic achievement [58]. Further studies should consider including these measures in order to analyze the effect of the intervention on other secondary outcomes, such as school performance, that may not be directly influenced by the intervention. Finally, this was a pilot study and the small number of students and schools involved highlights the need for a large-scale study. Whether these findings can be generalized is dependent on the next phase of the research, which is to compare the data from Italy and Australia. Doing so would enable an aggregation of a sufficient number of seriously bullied students to determine the impact of the P.E.A.C.E. pack intervention and if these promising preliminary findings can in fact be supported.

Notwithstanding these constraints, the present study offers interesting insights for school interventions focused on bullying phenomena. One point of strength is the effectiveness of our intervention for severe victims, that is, those who are commonly described as the most vulnerable and resistant to change [21]. Indeed, previous studies have described an improvement for severe victims, primarily with targeted intervention, including only seriously victimized students [23,24]. In contrast, in the present study, the intervention involved the whole class and was particularly effective for the victims. The second strength is the analysis of effectiveness based on the victimization before the intervention. In other words, our study suggests that an anti-bullying intervention can exert a different influence among students, opening new lines of research that could be followed. Another strength concerns the evaluation of the quality of the implementation process. This was foreseen in our research, as recommended by translational research [30], improving the value of the collected data and their interpretation.

## 5. Conclusions

It is well accepted that education is positively related to health, and that schools play a key role in promoting healthy behaviors and attitudes [59]. The responsibility of educators, as reported by The United Nations’ Convention on the Rights of the Child [60], is protecting children’s quality of life and their rights to be educated in a safe environment, free from all forms of violence, victimization, harassment, and neglect. Despite the considerable efforts of schools to address bullying issues, and international and national imperatives for schools to put in place policies and procedures to deal with the problem, much of the work undertaken in schools remains at the level of broadly based, and not necessarily research-informed interventions [61]. The findings from the present Italian study suggest that it is possible to effectively intervene to deal with bullying in schools, reducing severe victimization, increasing self-efficacy in bullied students, as well as their perception of classmates as more likely to intervene when a bullying episode occurs. Programs like the P.E.A.C.E. pack should be part of daily classroom activities, because they are effective and can be delivered entirely by teachers, representing a valid solution for schools with limited financial and time resources. In addition, our findings replicate those from Australian, Maltese and Greek research, opening new considerations on the effectiveness of the same intervention in different countries.

## Figures and Tables

**Figure 1 ijerph-17-05162-f001:**
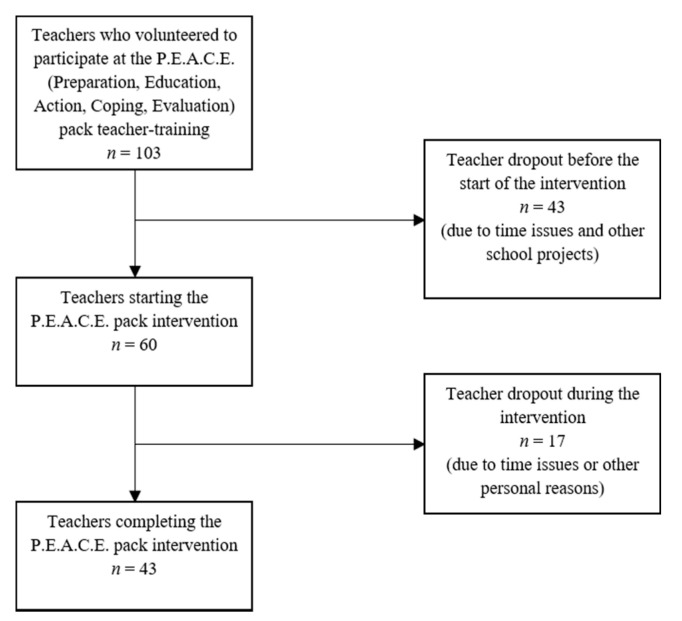
Flow diagram of the recruitment and selection of teachers.

**Table 1 ijerph-17-05162-t001:** Bullying victimization, self-efficacy and the perceived bystanders’ behavior of the not involved students.

Not Involved	Pre-Intervention			Post-Intervention			*Z*	*p*	*r*
	Mean	Standard Deviation	Median	Mean	Standard Deviation	Median			
Bullying Victimization	0.39	0.49	0	1.82	3.35	1	−7.620	<0.001	0.48
Self-efficacy	53.81	6.73	54	53.57	7.99	54	−0.049	0.961	0.01
Bystander Behavior (Students)	2.33	0.73	2	2.23	0.77	2	−1.980	0.048	0.13
Bystander Behavior (Teachers)	1.44	0.63	1	1.55	0.71	1	−2.585	0.010	0.16

Note: bullying victimization (*n* = 250); self-efficacy (*n* = 249); bystander behavior (students, *n* = 243); bystander behavior (teachers, *n* = 241).

**Table 2 ijerph-17-05162-t002:** Bullying victimization, self-efficacy and the perceived bystanders’ behavior of the occasional victims.

Occasional Victims	Pre-Intervention			Post-Intervention			*Z*	*p*	*r*
	Mean	Standard Deviation	Median	Mean	Standard Deviation	Median			
Bullying Victimization	3.21	1.01	3	3.56	3.76	3	−0.227	0.821	0.01
Self-efficacy	51.12	7.11	52	50.22	8.24	51.5	−1.254	0.210	0.08
Bystander Behavior (Students)	2.54	0.75	2	2.42	0.79	2	−1.970	0.049	0.13
Bystander Behavior (Teachers)	1.63	0.74	1	1.79	0.81	2	−2.553	0.011	0.17

Note: bullying victimization (*n* = 230); self-efficacy (*n* = 230); bystander behavior (students, *n* = 230); bystander behavior (teachers, *n* = 223).

**Table 3 ijerph-17-05162-t003:** Bullying victimization, self-efficacy and the perceived bystanders’ behavior of the severe victims.

Severe Victims	Pre-Intervention			Post-Intervention			*Z*	*p*	*r*
	Mean	Standard Deviation	Median	Mean	Standard Deviation	Median			
Bullying Victimization	8.60	2.78	8	7.48	6.20	6.5	−2.134	0.033	0.26
Self-efficacy	46.30	7.05	47	48.70	8.33	50	−3.194	0.001	0.38
Bystander Behavior (Students)	2.91	0.71	3	2.63	0.93	3	−2.530	0.011	0.30
Bystander Behavior (Teachers)	1.93	0.87	2	1.93	0.94	2	−0.015	0.988	0.001

Note: bullying victimization (*n* = 70); self-efficacy (*n* = 70); bystander behavior (students, *n* = 70); bystander behavior (teachers, *n* = 69).

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
