# Peer review of "The P.E.A.C.E. Pack Program in Italian High Schools: An Intervention for Victims of Bullying"

_ijerph, 2020, doi:10.3390/ijerph17145162_

Round 1

Reviewer 1 Report

Overall, I think this is a very valuable contribution in the field of preventing bullying. The only thing that could be clarified further is in the method section, since it was partly hard to follow (primarily the relation between 2.3 and 1.3 which discuss the program). Maybe, some parts of 2.3 should be integrated in 1.3 (e.g. presentation of the program), and in the method section have a clear focus on the participants, the questionnaire and the study design.

Author Response

Response to Reviewer 1 Comments

Point 1. Overall, I think this is a very valuable contribution in the field of preventing bullying. The only thing that could be clarified further is in the method section, since it was partly hard to follow (primarily the relation between 2.3 and 1.3 which discuss the program). Maybe, some parts of 2.3 should be integrated in 1.3 (e.g. presentation of the program), and in the method section have a clear focus on the participants, the questionnaire and the study design.

Response 1. We thank the Reviewer for appreciating our manuscript. We have rewritten paragraph 1.3. integrating some parts from paragraph 2.3. We also included more information in the method section regarding the participants, the questionnaire and the study design (see Response to Reviewer 2). All changes are highlighted in the manuscript.

Reviewer 2 Report

This is a well-written report on the PEACE pack program in Italian high schools. The paper make a significant contribution to the literature, theoretical background conducted is clearly explained and the contribution to research is properly elaborated. Also, the current manuscript has the potential to inform school settings in important ways. However, some recommendations are suggested in order to enhance the usefulness of this manuscript.

Introduction: Authors have included a good an extensive introduction and discussion with an overall rationale of the main variables reported into the study and how these variables fit with the obtained results. Good work

Method:

In the method setion is missing some important details.It would be great whether authors could indicate if a review protocol exist, if and where it can be accessed (if available).

Please, provide information about the code number of Institutional approval of the study. Also, it is suggested to provide more details about the process of obtaining consent from adolescents to participate in this study. For example, its recommended to explain how was consent obtained - written or verbal - and if parents had to consent for participation of underage subjects.

Include in the method section the number of participants who refused to participate as well as missing data. Also, how were student selected?

Elaborate on procedure (i.e. any compensation, did data collection occur during schools, were students recruited from specific classrooms, paper or online survey, etc.)

Explain inclusion and exclusion criteria. Also, explain the selection and rationale of variables to be included.

It is also important to include a flow diagram of the process of recruitment of sample.

Results:

Were there any covariates used in the statistical analyses, at least age, gender, SES, etc? Given that participants were not matched at baseline in intra-individual variables the control of potential confunders as age, sex… should be introduced in the model as a covariates to control sources of variability between groups.

The effectiveness of the effect is described using significance testing, where one should report effect sizes. So, please include effect size and explain the results in terms of effect size/ percentage of change across Time 1 (pre) to Time 2 (post)

Despite this limitations, the core of the article is important and relevant and I suggest the paper has potential after these changes.

Author Response

Response to Reviewer 2 Comments

Point 1: This is a well-written report on the PEACE pack program in Italian high schools. The paper make a significant contribution to the literature, theoretical background conducted is clearly explained and the contribution to research is properly elaborated. Also, the current manuscript has the potential to inform school settings in important ways. However, some recommendations are suggested in order to enhance the usefulness of this manuscript.

Response 1: We thank the reviewer for his/her positive comment. Below we provide a point-to-point description of all changes we made. All changes are highlighted in the manuscript.

Point 2: Authors have included a good an extensive introduction and discussion with an overall rationale of the main variables reported into the study and how these variables fit with the obtained results. Good work

Response 2: We thank the reviewer for appreciating the introduction.

Point 3: In the method section is missing some important details. It would be great whether authors could indicate if a review protocol exist, if and where it can be accessed (if available).

Response 3: Unfortunately, a review protocol does not exist. We will take this point into consideration for our future projects.

Point 4: Please, provide information about the code number of Institutional approval of the study. Also, it is suggested to provide more details about the process of obtaining consent from adolescents to participate in this study. For example, its recommended to explain how was consent obtained - written or verbal - and if parents had to consent for participation of underage subjects.

Response 4: We have entered all the information in the Method paragraph (paragraph 2.4.). In particular, we inserted the code number of Institutional approval. We also specified that parents had completed the written informed consent since the participants were minors. We also asked students for written informed consent, in line with the Declaration of Helsinki. 

Point 5: Include in the method section the number of participants who refused to participate as well as missing data. Also, how were student selected?

Response 5: We have inserted more information on the process of classes recruited and students’ selection (see flow diagram – Figure 1- and paragraph 2.1. Participants).

Point 6: Elaborate on procedure (i.e. any compensation, did data collection occur during schools, were students recruited from specific classrooms, paper or online survey, etc.).

Response 6: We have entered more information in the Method paragraph. In particular, we added the following information:

- No compensation was given for participation in the project (paragraph 2.4. Ethics);

- Data collection took place during school hours (paragraph 2.3. Procedure and study design);

- Questionnaires were completed online using computers in multimedia classrooms (paragraph 2.3. Procedure and study design);

- Information about classroom selected and student recruited (paragraph 2.1. Participants and flow diagram- Figure 1-).

Point 7: Explain inclusion and exclusion criteria. Also, explain the selection and rationale of variables to be included.

Response 7: We explained in the manuscript that teachers voluntarily participated in the study (paragraphs: 2.1. Participants and 2.3. Procedure and study design). No children were excluded due to SEN or difficulties in Italian language comprehension. Students could stop filling in the questionnaire at any time or leave questions empty (paragraph 2.4. Ethics).

Point 8: It is also important to include a flow diagram of the process of recruitment of sample.

Response 8: We included a flow diagram of the process of recruitment of teachers and classes (see Figure 1). We also described the flow diagram in the manuscript (paragraph 2.1. Participants).

Point 9: Were there any covariates used in the statistical analyses, at least age, gender, SES, etc? Given that participants were not matched at baseline in intra-individual variables the control of potential confunders as age, sex… should be introduced in the model as a covariates to control sources of variability between groups.

Response 9: We thank the Reviewer for his/her comments. We perform chi-square analyses in order to investigate the differences on gender and class levels among the three groups (severe victims, occasional victims and not involved) in the pre-intervention. No significant difference was found among the groups. We can include these analyses in the manuscript, if it could be relevant for the Reviewer. Unfortunately, since we used non-parametric analysis, we cannot include gender and class levels as covariates in the analysis to control sources of variability on the effect of the intervention between groups.

Point 10: The effectiveness of the effect is described using significance testing, where one should report effect sizes. So, please include effect size and explain the results in terms of effect size/ percentage of change across Time 1 (pre) to Time 2 (post)

Response 10: The effect sizes have been included in the manuscript. In detail, we included the description of the effect sizes in the paragraph 2.5 (Statistical analysis). We included the effect size values in the Results and in Table 1, 2, 3 (r values). We also included some considerations in the Discussion.

Point 11: Despite this limitation, the core of the article is important and relevant and I suggest the paper has potential after these changes.

Response 11: We thank the Reviewer for appreciating the relevance of our manuscript.

Reviewer 3 Report

This is an interesting article related to bullying interventions in high schools of Italy.

  1. Some sentences should be improved in terms of English grammar.

Example: 9th line of the Abstract

 “After the intervention severe victims…” to

“After the intervention, severe victims”

Please, read carefully the whole manuscript

  1. If data are available, a short paragraph should be added related to school performance of the victims before and after the intervention

  1. In the Discussion (page 9), among others, it is written “As already described in other contexts [9], when teachers are overloaded…”. Which is the personal opinion of the authors related to the “balance” between teacher’s intervention and spending hours to protect victims instead of improving their performance in terms of knowledge and teaching?

Author Response

Response to Reviewer 3 Comments

Point 1: This is an interesting article related to bullying interventions in high schools of Italy.

Response 1: We thank the Reviewer for appreciating our manuscript. All changes are highlighted in the manuscript.

Point 2: Some sentences should be improved in terms of English grammar.

Example: 9th line of the Abstract

 “After the intervention severe victims…” to

“After the intervention, severe victims”

Please, read carefully the whole manuscript

Response 2: The whole manuscript was carefully read by the two English mother-tongue co-authors (Slee and Skrypiec).

Point 3: If data are available, a short paragraph should be added related to school performance of the victims before and after the intervention

Response 3: Unfortunately, we did not collect data about school performance. We added this consideration as a limitation of our study, suggesting new directions for further studies (p. 11).

Point 4: In the Discussion (page 9), among others, it is written “As already described in other contexts [9], when teachers are overloaded…”. Which is the personal opinion of the authors related to the “balance” between teacher’s intervention and spending hours to protect victims instead of improving their performance in terms of knowledge and teaching?

Response 4: We agree with the Reviewer about the importance of addressing this theme. We added a few sentences to the discussion (p. 10).

Round 2

Reviewer 2 Report

I accept the current manuscrip in present form. Authors have done a well work and the topic is relevant. Congrats¡